# Has the Spread of African Swine Fever in the European Union Been Impacted by COVID-19 Pandemic?

**DOI:** 10.3390/ijerph19095360

**Published:** 2022-04-28

**Authors:** Vito Biondi, Salvatore Monti, Alessandra Landi, Michela Pugliese, Elena Zema, Annamaria Passantino

**Affiliations:** 1Department of Veterinary Sciences, University of Messina, 98168 Messina, Italy; vito.biondi@unime.it (V.B.); samonti@unime.it (S.M.); elena.zema96@gmail.com (E.Z.); passanna@unime.it (A.P.); 2Veterinary Practitioner, 98168 Messina, Italy; alelandi1612@gmail.com

**Keywords:** pigs, wild boars, pandemic, animal health, swine virus

## Abstract

African Swine Fever (ASF) is a contagious viral disease of domestic and wild pigs, listed as notifiable by the World Organization for Animal Health (OIE). It causes substantial economic losses to pig farming in the affected countries, with consequent enormous damage to livestock production due to mortality of the animals, and to the restrictions on national and international trade in pigs and derivative products that the presence of the infection implies. To prevent or reduce the risk of ASF introduction, the World Trade Organization (WTO) and the OIE recommend preventive and control measures, such as the ban of live swine and their products traded from ASF-affected to ASF-free countries or zones. The current spread of ASF into Europe poses a serious risk to the industrialized and small-scale pig sector, as demonstrated by observed cases in different EU areas. In this paper the authors discuss the impact of the COVID-19 pandemic on ASF, and the indirect effects including the impact on animal health and disease management. They suggest that the COVID-19 pandemic has severely affected animal disease surveillance control. ASF requires rapid responses and continuous monitoring to identify outbreaks and prevent their spread, and both aspects may have been greatly reduced during the COVID-19 pandemic.

## 1. Introduction

African Swine Fever (ASF) is a contagious hemorrhagic viral disease of domestic and wild swine, causing substantial economic losses to pig farming in the affected countries. ASF can be transmitted either via direct animal contact with infected animals (wild or domestic swine), via ingestion of contaminated food (e.g., sausages or uncooked meat, swill feed), or through indirect contact with fomites (vehicles, equipment, clothes, footwear, etc.) or soft ticks bites [1]. The movement of infected animals, contaminated pig products, and the illegal disposal of carcasses are the most significant means of the disease spread. Due to its high mortality, ASF is a notifiable disease included in the list reported by the World Organization for Animal Health (OIE), which includes all diseases representing a danger to animal and public health. Although ASF does not pose any health threats for humans, its consequences on the swine industry are catastrophic. It is considered one of the most devasting diseases in pigs and wild boar. In fact, this disease, against which there is not currently an effective treatment or vaccine, can cause enormous damage to pig livestock production: both directly due to mortality, and indirectly due to the restrictions on the trade in live animals and their products, both for the internal market and for international trade with third countries that the presence of the infection implies, with the associated depreciation of their pig products. The spread of ASF has internationally negative impacts on animal health and welfare, provoking socio-economic impacts on livelihoods, national food security, and for international markets and trade, and therefore has significant potential to impede the coordinated efforts to reduce hunger and poverty worldwide under the Sustainable Development Goals.

Early detection, prevention, and reporting are crucial to controlling the disease. The prevention in countries free of the disease depends on: the application of appropriate import policies (ensuring that neither infected live pigs nor pork products are introduced into areas free of ASF); strict biosecurity measures (frequently cleaned and disinfected farms, transport vehicles); and improved husbandry practices and production systems.

The current spread of ASF in the European Union (EU) and its recent incursion into some areas of Italy, Poland and Slovakia constitutes a serious risk to EU pig producers. 

To combat the risks associated with the spread of the disease in a proactive manner, EU areas affected by this disease are listed as restricted zones. Based on the current epidemiological situation of ASF in the Union, new restricted zones of a sufficient size are defined, such as for Poland and Slovakia, as laid down in Regulation (EU) 2022/136 [2] and Regulation (EU) 2022/205 [3]. Given that the situation of ASF is very dynamic in the Union, the European legislator has taken account of the situation in the surrounding areas when establishing these new restricted zones.

Based on the current situation in EU countries, the objective of this paper was to analyze if and how the COVID-19 pandemic has had an indirect impact on the spread of ASF in the EU.

## 2. Current Status of African Swine Fever during COVID-19 Pandemic in EU

ASF is not considered a new disease; in fact, it has been around for decades and has made its way to several continents, including Europe [4]. In 1978, the disease was introduced to the Italian island of Sardinia and has since become endemic [5,6]. It spread from Africa to Georgia in 2007, and has since spread throughout the Caucasus, Russia and Eastern Europe, all the way to Central Europe and even Belgium [7]. During recent years (2015–2020), ASF cases were reported in several EU member states (Belgium, Bulgaria, Czech Republic, Estonia, Germany, Greece, Hungary, Sardinia in Italy, Latvia, Lithuania, Poland, Romania, and Slovakia) [8]. In all EU countries, except for Romania, wild boar was the main affected species [9].

During the COVID-19 pandemic, which broke out in December 2019, an increased incidence of African swine fever cases was observed. A summary of the ASF situation by world region during the COVID-19 pandemic (2020–2022) is reported in Table 1.

The EU area affected by ASF has progressively expanded since 2019 and so far, several countries continue to be affected. The EU Animal Disease Information System (ADIS) that is administered by the European Commission documents the situation of important infectious animal diseases in each EU country, and shows that ASF cases in wild boar specifically have been on a steep increase.

Data demonstrate that during 2020 in Europe 11,027 infected wild boar were found [11], with a 41.89% increase compared to what was found in the whole of 2019 (6407 cases) [12]. More recently (from 1 January 2022, until 13 February 2022), according to ASID, there were 1572 outbreaks among wild boar and 81 among domestic pigs in the EU [13]. Cases in wild boars have been registered in 11 countries as shown in Figure 1.

The most outbreaks were observed in Poland (521), followed by Germany (254), Bulgaria (201), and Romania (180). Additionally, cases were registered in January in Bulgaria, Estonia, Hungary, Italy, Latvia, Lithuania, Slovakia, and Ukraine. Relating to outbreaks among domestic pigs, Romania accounts for 63, Serbia for 11, Bulgaria for 2, and Republic of North Macedonia, Italy, Moldova, Slovakia and Ukraine for 1 each.

Relating to outbreaks of ASF in domestic pigs, cases have not increased (Figure 2).

The situation varies between the member states, due to the structure of domestic pig production (in particular, the proportion of backyard holdings), and also geographical conditions and the characteristics of the wild boar population.

Although Italy was free from ASF, excluding the island of Sardinia where the disease (ASF genotype 1) has been endemic since 1978 as abovementioned, new outbreaks have recently been reported. At the start of January 2022, the first case of ASF genotype 2 was found in a wild boar carcass in the Municipality of Ovada (Alessandria province, Piedmont Region) in northern Italy. That case was confirmed by the National Reference Center for Swine Fever of the Experimental Zooprophylactic Institute of Umbria and Marche. Further cases were found in neighboring areas, in the province of Genoa (Liguria Region) and in Alessandria (Piedmont Region). Accordingly, the Ministry invited regions of Piedmont and Liguria to suspend hunting in all municipalities falling within the infected area, and it gave regions 30 days to implement a series of measures to contain and eradicate the disease. In the southern region of Campania, preparations are also underway for the protection of native breeds such as black Casertana pigs.

This situation confirms that ASF continues to spread in several countries with serious impacts on the pig production system, animal health and welfare, as well as on national food safety and international trade.

## 3. Discussion

Pulmonary infection (COVID-19) caused by the new severe acute respiratory syndrome coronavirus 2 (SARS-CoV-2), which emerged with the first cases in China in late 2019 and was declared a global pandemic on 11 March 2020 by the World Health Organization’, has disrupted the social fabric and lifestyle causing severe repercussions on health systems and the global economy. However, the COVID-19 pandemic may have affected several aspects of the entry pathways for ASFV into ASF-free countries. Since December 2019, ASF has continued to spread to new areas because of the diffusive nature of the virus. This further spread may have been facilitated by difficulties in animal disease prevention and control due to the various lockdowns of COVID-19. Measures taken to mitigate the pandemic have restricted the movement of people, causing an indirect economic crisis. The latter, exacerbated by a high demand for laboratory supplies for human COVID-19 diagnosis and research, probably also affected the ability of veterinary laboratories to fulfill their mandates. In addition, official international reporting of the ASF may have been disrupted due to logistical issues and constraints faced by countries. Timely and accurate reporting allows the prevention of ASFV entry from affected areas. Thus, a lack of knowledge about the health status of animals in some areas may have increased the risk of virus entry through legally imported swine products/subproducts. In contrast, the risk of entry should be reduced considering that the COVID-19 pandemic negatively affected the global meat trade with a decline in meat imports and a decrease in meat prices. Similarly, the risk of ASFV entry through smuggling pork in passenger luggage may have decreased due to the reduction in the number of people traveling. However, the number of import/export inspectors has decreased worldwide, so border controls may have been compromised. Restrictions at ports and borders, curfews, and limitations on social distance have led to reduced quality, productivity, and competitiveness in key productive sectors. Restrictions have hit the livestock sector hard by disrupting the feed supply chain, reducing livestock services, limiting animal health services including delays in disease diagnosis and treatment, limiting access to markets and consumers, and reducing labor force participation. In summary, the closures and restrictions on local and international trade triggered by the COVID-19 have affected food production, animal production, and animal health and welfare [14,15].

ASF is a major crisis due to its high mortality rate, affecting rural and swine industries worldwide. Therefore, in 2019, the Food and Agriculture Organization (FAO) and OIE declared ASF a global priority. As a result, these entities included ASF in the Global Framework for the Progressive Control of Transboundary Animal Diseases (GF-TAD) to resist massive disease burdens, especially at the onset of the COVID-19 pandemic. ASF has had a significant impact on affected countries’ livestock producers, communities, and economies. At the same time, it induces trade restrictions due to the lack of approved treatments or vaccines. As a result of this restriction, the depopulation of all affected and exposed pig farms remains the only means of preventing the disease. During the pandemic, restrictive regulations and closures have worsened trade and consumption in the swine industries. Many industries, particularly those related to pharmaceutical manufacturing and grain markets, were adversely affected by ASF disease even before the emergence of COVID-19. However, the trade and day-to-day operations of these sectors deteriorated during the COVID-19 pandemic for several reasons: global efforts to prevent the disease postponed more than 30% of their field activities and measures; the ability of veterinary services to detect the disease early decreased because of movement and trade restrictions between regions; and the focus of many health care professionals shifted away from PSA to help combat COVID-19. Thus, fewer precautions and protective measures have been employed and adhered to.

Additional emerging infectious disease outbreaks are a significant concern, as the medical, diagnostic, and supply infrastructure has been severely stressed by the urgent needs of these two pandemics. National veterinary and animal health services have assisted in the diagnosis of SARS-CoV-2, severely limiting the ability to diagnose zoonotic and endemic diseases in animals. Thus, an outbreak may go undiagnosed or be underdiagnosed, hampering control efforts. Highlighting the reality of this risk, avian influenza outbreaks were reported in Australia, Taiwan, Hungary, Poland, and the United States during COVID-19. In the literature, it has been reported that changes in human behavior during the pandemic led to a record number of salmonella outbreaks (from backyard chicken farming) and a fear of increased cases of Lyme disease (attributed to increased outdoor activities amidst a climate pattern that favors tick populations) [16]. Social detachment and hygiene behavior dictated by COVID-19 could increase awareness of the need for safe practices, and lead to the observation of higher levels of biosecurity in livestock production systems where required.

Limiting movement, an effective measure to prevent transmission of COVID-19, affected activities related to animal health and disease management of farm animals. Some of the veterinary activities, such as prophylactic disease controls, provided their services by working at a reduced intensity, or were suppressed during the pandemic. This, combined with increased wildlife–livestock contacts and longer livestock stays on farms, resulted in an impact on the spread and incidence of communicable animal diseases [17]. In fact, the long-term effects of COVID-19 on animal health will be strongly influenced by the impact of the crisis on farmers’ livelihoods and the capacity of animal health services [18,19]. ASF and COVID-19 are strong examples of the need to apply a One Health approach to disease control, as they demonstrate the devastating results of the contagious spread of a virulent infection across a significant portion of the globe due to animal, human, and environmental interactions [20]. This notion emphasizes that pathogen ecology and disease management must consider human, animal, and environmental perspectives, implying that physicians, veterinarians and ecologists should work together to effectively manage health issues. According to the OIE, veterinarians are important members of the global health community and play an important role in disease prevention and management, particularly for diseases that are communicable to humans. In addition, veterinary laboratories are currently dedicated to COVID-19 under the supervision of health authorities. Unfortunately, vital measures to control COVID-19 have had the negative trade-off of jeopardizing animal disease elimination and prevention activities. International organizations and multiple researchers have raised concerns about the potential downstream impacts of the COVID-19 pandemic on the control of the human, animal, and neglected tropical diseases [21,22].

## 4. Conclusions

The COVID-19 pandemic has inevitably had a lasting impact on both people and animals. Although it is too early for a full assessment, we suggest that the COVID-19 pandemic and the subsequent economic crisis will severely affect animal health in several ways [23]. In the short term, some of the veterinary activities regarded as essential by the OIE, such as preventative measures against diseases with significant public health or economic impact, are already either working at a lower intensity or have been suppressed during the lockdown. This, in addition to other short-term effects such as increased wildlife–livestock contacts, less population control, or longer on-farm stays of stock, will trigger effects on the distribution and incidence of transmissible animal diseases [17,24]. A more comprehensive analysis of the phylogeography and evolutive dynamics of ASFV genotypes in the endemic areas may be helpful in the disease control. The different diffusion of two genotypes and the virus’s adaption to new environments following the transfer out of Africa to Eurasia have assumed global importance in the diffusion of disease. However, the ASFV transmission is strongly influenced by the genotype, and an important role in the diffusion of the disease is played by the Ornithodoros tick vector for genotype I, and infection of wild boar populations for genotype II [25].

In conclusion, this study aims to highlight the strong connection between ASFV-affected countries in Europe during the COVID-19 pandemic. The results presented are not intended to provide an estimate of ASFV risk, but they highlight important aspects to consider during a global public health emergency. The long-term consequences of this pandemic on countries’ abilities to control and prevent ASFV are still unclear. However, it remains critical to ensure that the risk of ASF is not underestimated. Finally, an increased surveillance effort at the borders of each European country is needed to ensure early detection of ASFV and rapid response.

## Figures and Tables

**Figure 1 ijerph-19-05360-f001:**
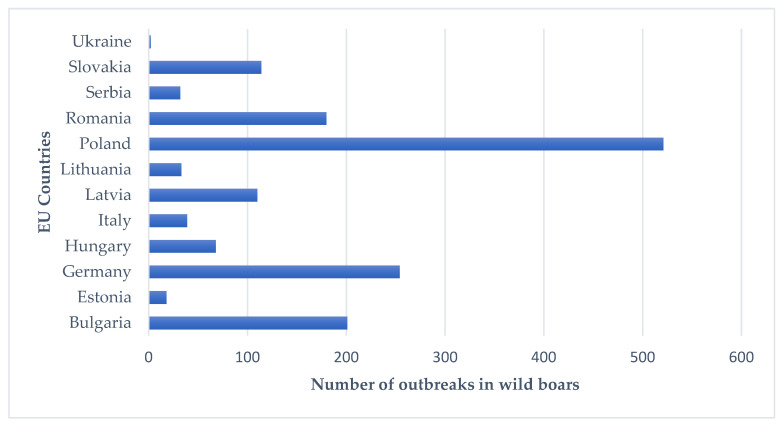
Number of outbreaks for ASF in wild boars in European countries in 2022.

**Figure 2 ijerph-19-05360-f002:**
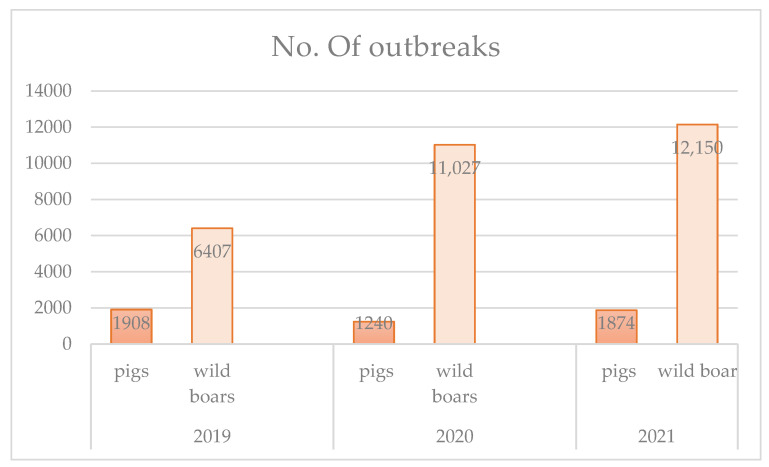
Number of outbreaks of ASF in domestic pigs and wild boars in European countries in the period 2019–2021.

**Table 1 ijerph-19-05360-t001:** Data of the outbreaks number, cases, and animal losses caused by ASF in the different world regions since January 2020 [10].

World Regions	Outbreaks		Cases		Losses ^(1)^
	Domestic Pigs	Wild Boar	Domestic Pigs	Wild Boar	Domestic Pigs
Africa	149		12,626		19,970
Americas	210		8592		14,972
Asia	1039	1518	89,035	1625	398,247
Europe	3336	16,258	928,376	27,672	1,260,551
Oceania	4		500		397
**Total**	**4738**	**17,776**	**1,039,129**	**29,297**	**1,694,137**

^(1)^ The impact of ASF has been measured in terms of losses in the establishments affected by the outbreaks, and includes animal deaths, animals killed, and their disposal. It does not include animals culled in areas around the outbreak to control the disease.

## Data Availability

Data are available, contacting the corresponding author (michela.pugliese@unime.it).

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
