# Peer review of "Has the Spread of African Swine Fever in the European Union Been Impacted by COVID-19 Pandemic?"

_ijerph, 2022, doi:10.3390/ijerph19095360_

Round 1
Reviewer 1 Report
In the manuscript " Has the spread of the African Swine Fever in European Union been impacted by COVID-19 pandemic?" the authors analyze the situation of the AFS diffusion respect to Covid-19 diffusione and outbreak. The article could be the interest for the reader of the Journal and I suggest to publish it after major revision. In particular:
1. The article is poor in data (table , figures ecc,) that allow to better understand the situation of the ASF diffusiona and how the covid outbreak has incresae this situation.Please, the authors should improve it.
2. The table 1 leggend is poor in details. Please, the authors should provide it.
3 The figure 1 lacks both axis names. Please, the authors should provide it.
4. The authors mentioned that the diffusion of two different ASF genotypes. How is the distribution in the studied regions of these genotypes? Is there a prevalence of one respect the other? Please the authors should add this information.
Author Response
Dear Reviewer,
Thank you very much for your time and all your comments.
We have revised the manuscript considering your comments; the answers to your questions are given below.
The changes made in the manuscript to address comments are marked up using the
“Track Changes” function.
In the manuscript " Has the spread of the African Swine Fever in European Union been impacted by COVID-19 pandemic?" the authors analyze the situation of the AFS diffusion respect to Covid-19 diffusione and outbreak. The article could be the interest for the reader of the Journal and I suggest to publish it after major revision. In particular:
- The article is poor in data (table , figures ecc,) that allow to better understand the situation of the ASF diffusiona and how the covid outbreak has incresae this situation.Please, the authors should improve it.
R. We have added a figure.
- The table 1 leggend is poor in details. Please, the authors should provide it.
R. We have provided.
3 The figure 1 lacks both axis names. Please, the authors should provide it.
R. We have provided that.
- The authors mentioned that the diffusion of two different ASF genotypes. How is the distribution in the studied regions of these genotypes? Is there a prevalence of one respect the other? Please the authors should add this information.
R. Genotypes are used to trace the origin of ASFV during outbreaks. 22 different p72 (protein 72) genotypes have been identified from East and South African countries. Genotype 1 is predominant in West Africa. Outside Africa, genotype I was the only one found in Europe, America, and the Caribbean, until the introduction of genotype II in 2007 into Georgia 2007 from East Africa. Current available molecular data derived by using standardized genotyping procedures have indicated the presence of the sole p72 genotype II circulating in eastern European, thus indicating a single introduction in 2007.
Western and Central Africa have traditionally shown the presence of genotype I sequence with low genetic variability and an absence of the sylvatic cycle, although transfer and dissemination of ASFV genotypes from eastern to western Africa have been demonstrated.

Reviewer 2 Report
The paper giving an overview of what is the
problem related to ASFV. Authors analyzed the problem of the disease that has been present in Italy since 1978, but which has recently had a surge mainly linked to the repercussions of the Covid 19 pandemic.
Analysis shows that veterinary laboratories have been unable to carry
out screening related to ASFV disease due to the Covid 19 pandemic.
The conclusions underline the probable strong connection between
ASFV-european effected countries and Covid 19 pandemic with long term
consequences.
the matter about ASFV in Italy and in Europe is of considerable interest but I think that the conclusions shown are reductive. Authors should point out better and should be investigate the matter with more accurate scientific proposals and suggestion.
Author Response
Dear Reviewer,
Thank you very much for your time and all your comments.
We have revised the manuscript considering your comments; the answers to your questions are given below.
The changes made in the manuscript to address comments are marked up using the
“Track Changes” function.
Conclusions have been improved as required.

Round 2
Reviewer 1 Report
Thank's the authors for the new version of the manuscript. Now, it is ready to be published.
Reviewer 2 Report
The paper is better after revision and my suggestion is accept in present form.